# A Polysaccharide-RBD-Fc-Conjugated COVID-19 Vaccine, SCTV01A, Showed High Immunogenicity and Low Toxicity in Animal Models

**DOI:** 10.3390/vaccines11030526

**Published:** 2023-02-23

**Authors:** Chunyun Sun, Desheng Kong, Erhong Guo, Jun Zhao, Jilei Jia, Rui Wang, Juan Ma, Meng Chen, Jianbo Lu, Chulin Yu, Kuokuo Li, Liangzhi Xie

**Affiliations:** 1Beijing Engineering Research Center of Protein and Antibody, Sinocelltech Ltd., Beijing 100176, China; 2Beijing Key Laboratory of Monoclonal Antibody Research and Development, Sino Biological Inc., Beijing 100176, China; 3Cell Culture Engineering Center, Chinese Academy of Medical Sciences & Peking Union Medical College, Beijing 100005, China

**Keywords:** COVID-19, SARS-CoV-2, conjugated vaccine, immunogenicity, toxicity

## Abstract

We previously developed a polysaccharide-–RBD-conjugated nanoparticle vaccine which induced protective efficacy against SARS-CoV-2 in a mouse model. Here, we newly developed a vaccine, SCTV01A, by chemically conjugating recombinant SARS-CoV-2 RBD-Fc and PPS14 (*Streptococcus pneumoniae* serotype type 14 capsular polysaccharide). The immunogenicity and toxicity of SCTV01A were evaluated in animal models. The PPS14 conjugation enhanced the immunogenicity of RBD-Fc in C57BL/6 mice whether formulated with SCT-VA02B or Alum adjuvant. SCTV01A also induced high opsonophagocytic activity (OPA) against *S. pneumoniae* serotype 14. In addition, SCTV01A stimulated potent neutralizing titers in rhesus macaques and effectively reduced lung inflammation after SARS-CoV-2 infection with neither antibody-dependent enhancement (ADE) nor vaccine-enhanced diseases (VED) phenomenon. Importantly, the long-term toxicity study of SCTV01A in rhesus macaques did not cause any abnormal toxicity and was tolerated at the highest tested dose (120 μg). The existing immunogenicity and toxicological evaluation results have demonstrated the safety and efficacy of SCTV01A, which will be a promising and feasible vaccine to protect against SARS-CoV-2 infection.

## 1. Introduction

Severe acute respiratory syndrome coronavirus 2 (SARS-CoV-2) led to coronavirus disease 2019 (COVID-19) and has become a pandemic disease and caused more than 6 million deaths worldwide [WHO]. Due to the severity of the past few years, it is necessary to design high-efficiency vaccines against COVID-19 to slow down and inhibit its global spread. Based on the different targets and technologies, vaccines can be divided into five categories: nucleic acids (RNA and DNA) vaccines, viruses (live attenuated and inactivated) vaccines, recombinant protein vaccines, viral vectors (non-replicating and replicating) vaccines, and virus-like particle vaccines [1]. Currently, the World Health Organization (WHO) announced that 172 vaccines have entered clinical studies worldwide (including 47 vaccines approved to be marketed or received emergency use authorization) and 198 vaccines are under preclinical studies [2].

The SARS-CoV-2 receptor-binding domain (RBD) is the protruding site of the Spike (S) protein that mediates viral-cell fusion during the initial infection event through binding of the human receptor angiotensin-converting enzyme 2 (hACE2) [3,4]. Analysis of antibodies in the plasma of COVID-19 patients showed that most patients had produced antibodies against receptor binding domain (RBD) around day 10 after the onset of symptoms [5]. Cellular immunity analysis in patients showed that RBD-specific T cell responses in patients newly diagnosed with COVID-19 significantly elevated when compared with healthy people. Preclinical studies found that RBD protein can induce both humoral immunity and T cell immune response in animals and exhibit protection against SARS-CoV-2 challenge [6,7,8]. Furthermore, the RBD contains multiple neutralizing epitopes, several RBD-binding monoclonal antibodies (mAbs), such as REGN10933, REGN10987 [9], HB27 [10], H89Y [11], S309 [12], SA58, and SA55 [13], exhibit great neutralizing potency or cross-neutralizing activity against SARS-CoV-2 and SARS-CoV. Therefore, RBD can be considered as an effective target of the SARS-CoV-2 vaccine. Several vaccines based on RBD have been approved for marketing, showing good immunogenicity and protective effects in naïve and boost populations.

Fcγ receptors (FcγRs) belong to the immunoglobulin superfamily, can trigger the activation of innate effector cells, and play important roles in antigen presentation, maturation of dendritic cells (DC), B cell activation, and plasma cell survival [14,15,16]. When antigens are linked to the Fc region of an immunoglobulin, antigen processing and presentation can be improved through Fc-FcγR-mediated antigen targeting to antigen presenting cells (APCs) as well as subsequent endocytosis [17]. Furthermore, the Fc-fusion can also improve the stability of recombinant immunogens and extend their in vivo half-life after injection by interacting with human neonatal Fcγ-receptor (hFcRn) [18]. Thus, the Fc-fusion antigens are also called “second generation” subunit vaccines [19].

Capsular polysaccharides (PS) are acidic polysaccharides made up of repeating oligosaccharide units, which may be either linear or branched [20,21]. When a protein antigen is covalently conjugated to PS, the PS-mediated antigen crosslinking can induce stronger B-cell receptors (BCRs) cross-linking, antigen internalization, as well as presentation by APCs [22,23]. Meanwhile, PS can bind to C-type lectin receptors (CLR) on APCs, resulting in antigen “delivery” to APCs. Several Pneumococcal polysaccharides vaccines (PPSV), such as PCV13 and PPSV23, are on the market for prevention of infection caused by pneumococcal bacteria. Among these *S. pneumoniae* pneumococcal polysaccharides (PPS), PPS from *S. pneumoniae* serotype 14 (PPS14) showed a superior binding capacity to CLRs on DCs [24,25,26] and can be efficiently internalized by DCs [27]. The contribution of PPS14 to enhance the immunogenicity of RBD has been verified in our previously developed nanoparticle vaccine which chemically conjugates recombinant SARS-CoV-2 RBD and PPS14 [28].

In this study, a novel nanoparticle vaccine, SCTV01A, was designed and prepared by chemically conjugating recombinant SARS-CoV-2 RBD-Fc and PPS14. SCTV01A induced higher SARS-CoV-2 RBD-specific neutralizing antibodies (nAbs) than RBD-Fc and protected nonhuman primates against SARS-CoV-2 infection. Moreover, SCTV01A showed good tolerance and did not cause any abnormal toxicity in nonhuman primates. Therefore, SCTV01A may be a safe and effective vaccine to protect against SARS-CoV-2 and *S. pneumoniae* infection.

## 2. Materials and Methods

### 2.1. Animals

Female C57BL/6 mice at 8 weeks of age were purchased from Beijing Vital River Laboratory Animal Technology, China. The animals were housed in a specific pathogen-free facility. Rhesus macaques at 5–9 years old for challenge study were provided by the Institute of Medical Biology, Chinese Academy of Medical Sciences. Two days before the challenge, the animals were transferred to the P4 large animal laboratory. Rhesus macaques at 4–6 years old for long-term toxicity study were obtained from Guangxi Grandforest Scientific Primate Company, Ltd., China. All animal experiments were carried out following biosafety operating regulations and animal ethics, and humane care was given to animals to ensure animal welfare.

### 2.2. Reagents

Specific reagents were listed: Luciferase Assay System (E1501) and Passive Lysis 5× Buffer (E1941) were purchased from Promega (Madison, WI, USA). PMA (P1585), Ionomycin (407953), paraformaldehyde, Gelatin (48722-500G-F), and N, N-Dimethylformamide (DMF) (327171000) were purchased from Sigma-Aldrich (Saint Louis, MO, USA). eBioscience™ Brefeldin A Solution (1000×) (00-4506-51) was purchased from Invitrogen (Carlsbad, CA, USA). Rabbit F(ab’)2 anti-Mouse IgG (F(ab’)2)-HRP (315-036-047) was purchased from Jackson ImmunoResearch (West Grove, PA, USA). HRP conjugated polyclonal Goat against monkey IgG (ab112767) was from Abcam (Cambridge, MA, USA). Mouse IFN-gamma ELISpot^PLUS^ (ALP), strips (3321-4AST-2), and Mouse IL-4 ELISpot^PLUS^ (ALP) (3311-4APW-2) were purchased from Mabtech (Nacka, Sweden). RBD peptides (15-mer peptides overlapping by 11 amino acids) were synthesized by Scilight-Peptide (Beijing, China). Recombinant RBD-his (40592-V08H111) and RBD protein (40592-VNAH) were purchased from Sino Biogical Inc. (Beijing, China). Pneumococcal Polysaccharide Powder Type 14 (81-X^TM^) was purchased from ATCC (Rockville, MD, USA). FBS, 10× HBSS (without Ca^2+^, Mg^2+^, Phenol Red, 14185-052), 10× HBSS (with Ca^2+^, Mg^2+^, without Phenol Red, 14065-056), and 100× GlutaMax-1 (35050-061) were purchased from Gibco (New York, NY, USA). THB Broth (PN01216) was purchased from Shanghai Canspec Scientific & Technology Co. (Shanghai, China). Bacto Agar (214010) and Bacto Yeast Extract (212750) were purchased from Becton-Dickinson (Heidelberg, Germany). Baby Rabbit Complement (31064-5) was purchased from Pel-Freez Biologicals (Rogers, AR, USA). Cell lines included: HEK293T, Vero E6, HL60, and CHO cells were purchased from ATCC; 293FT-ACE2 cells from SinocellTech; and Huh-7 cells from CCTCC, China. Mono-clone *S. pneumoniae* strain was purchased from National Institutes for Food and Drug Control of China. SCT-VA02B and Alum adjuvants were manufactured by SinoCellTech (Beijing, China).

### 2.3. Plasmid Construction and Expression of RBD-Fc

The DNA fragment for the expression of RBD (RBD amino acid 319 to 541) of Wuhan-Hu-1 was amplified by PCR using a 669-bp full-length sequence as the template and fused with human IgG1 Fc (RBD-Fc). The amplified DNA fragment was cloned into the plasmid of pc3-L1+SV40-R at sites of *Kpn* I and *Not* I. The recombinant plasmid was characterized by DNA sequencing. Subsequently, the plasmid was transfected into Chinese hamster ovary (CHO) cells and following glutamine synthetase (*GS*)-based selection, the RBD-Fc stably expressing CHO cell line was established for producing RBD-Fc protein.

### 2.4. Preparation of Pneumococcal Capsular Polysaccharides

*S. pneumoniae* isolate CMCC 31614 (serotype 14) was obtained from the National Center for Medical Culture Collections (CMCC, China). *S. pneumoniae* were cultured and lysed, and pneumococcal capsular polysaccharides were prepared as previously described [28].

### 2.5. Conjugation of RBD-Fc to PPS14

The conjugation procedures were conducted by reductive amination method as previously described [29]. Briefly, PPS14 was oxidized with NaIO_4_ and then mixed with RBD-Fc in Na_2_HPO_4_ buffer and reacted with sodium cyanoborohydride solution in rotation for 16 h at room temperature (RT). The mixture was further treated with sodium borohydride solution to reduce any remaining aldehyde. The conjugate samples were obtained by ultra-filtration with a cut-off molecule of 100,000 MW.

### 2.6. Reduced Polyacrylamide Gel Electrophoresis

Using the Novex NuPAGE SDS-PAGE (NuPAGE) gel system, the recombinant RBD-Fc protein (5 μg) was characterized by polyacrylamide gel electrophoresis on gradient gels. The samples were tested in triplicate at 200V constant voltage for 35 min. The gels were then stained with Coomassie brilliant blue, and the protein bands were analyzed by a gel imaging system.

### 2.7. Size Exclusion Chromatography Coupled with Multi Angle Light Scattering (SEC-MALS)

The purity and average molecular weight of SCTV01A were analyzed by high-performance liquid chromatography (HPLC) (model: Agilent 1260) with an analytic TSK gel G3000 SWxL column (7.8 × 300 mm, 5 μm) and a MALS detector (Wyatt Technology Corp. Santa Barbara, CA, USA). The buffer (200 mM disodium hydrogen phosphate, 100 mM arginine, 1% isopropanol, pH 6.5) was used as the mobile phase with 280 nm ultraviolet detection at a flow rate of 0.5 mL/min and a column temperature of 25 °C.

### 2.8. Transmission Electron Microscopy (TEM)

The generated SCTV01A nanoparticle vaccine was characterized by TEM. Briefly, the generated SCTV01A conjugated proteins (0.3 mg/mL) were loaded on the copper net. The samples were stained with phosphotungstic acid after being dried. Transmission electron microscope (Hitachi, model: H-7650) was used to observe and photoimage the proteins.

### 2.9. Dynamic Light Scattering (DLS)

SCTV01A nanoparticle vaccine was equilibrated at 25 °C and particle size distribution was analyzed using Dynamic light scattering (DLS) (Wyatt Technology Inc, DynaPro NanoStar, Santa Barbara, CA, USA). The data was analyzed by Dynamics 7.1.8 software.

### 2.10. Receptor Binding of SCTV01A to Human ACE2 by BLI

Bio-Layer Interferometry (BLI, Octet RED96e, Fortebio) was used to analyze the binding affinity and kinetics of recombinant RBD-Fc and SCTV01A to biotinylated human ACE2-His performed by PALL Corporation’s biomolecular interaction instrument. Streptavidin probes were used to capture biotinylated ACE2-His (5 μg/mL) and different concentrations of recombinant RBD-Fc or SCTV01A protein for a reaction time of 180 s and dissociation time of 180 s. The affinity constant was analyzed by data analysis software.

### 2.11. Immunogenicity Analysis of SCTV01A in Mice

Female C57BL/6 mice (6–8 weeks old, n = 6/group) were intramuscularly immunized with 1 μg or 3 μg of SCTV01A formulated with 2 mg SCT-VA02B or 1 mg Alum (with a volume ratio of 1:1) and an equal volume of adjuvant alone used as the negative control for three times with an interval of 14 days. Animals were bled from the tail veins, followed by RBD-specific IgG titer and neutralizing titer detection. At thirty-three days post the third immunization, these animals were euthanized and their spleens were dissected for enzyme-linked immunospot (ELISpot).

### 2.12. RBD-Specific Antibody ELISA Assay

Anti-RBD antibody in mouse serum was detected by ELISA. Briefly, 5 μg/mL of SARS-CoV-2 RBD-his protein was coated into ELISA plates overnight at 2~8 °C. After blocking with 2% BSA-TBST buffer, 100 μL of diluted serum samples were transferred to the plates and incubated for 1 h. Rabbit F(ab’)2 anti-Mouse IgG (F(ab’)2)-HRP secondary antibody (1:5000) was added and incubated at room temperature for 1 h. For monkey serum samples, recombinant RBD was diluted to 2 μg/mL, added into the plate, and incubated at 2~8 °C overnight. After washing 3 times, 2% casein-PBST blocking solution was added and incubated for 1 h at room temperature. Serum samples diluted with 0.1% casein-PBST at the ratio of 1:1000 were further diluted with 1% mixed unimmunized serum and added 100 μL/well to the plate, followed by an incubation at room temperature for 2 h with shock. HRP conjugated polyclonal Goat against monkey IgG was diluted to 15 ng/mL with 0.5% casein-PBST, added to the plate, and incubated at room temperature in the dark for 1 h. Plates were washed 5 times, and color development reaction was carried out by adding TMB substrates, and stopped by 2 M H_2_SO_4_. Optical absorbance at 450 nm (OD_450_ nm) was read in a microplate reader (BioTek). Antibody titers were defined as the highest serum dilution ratio that resulted in OD_450_ nm 2.1-fold higher than background values from serum samples of unimmunized mice.

### 2.13. SARS-CoV-2 Pseudovirus-Based Neutralization Assay

Pseudoviruses used in neutralization assays were constructed as described previously [28]. Pseudoviruses of Wuhan-Hu-1, Delta, BA.1, and BA.4/BA.5 were constructed based on GISAID accession ID EPI_ISL_402125, EPI_ISL_1999775, EPI_ISL_6640917, and EPI_ISL_11542270.1, respectively. For pseudovirus neutralization assay, vaccine immunized serum samples (heat-inactivated at 56 °C for 30 min) were serially diluted, incubated with 100 TCID_50_/well pseudovirus (1 h at 37 °C, in a 5% CO_2_ incubator), and co-cultured with 3 × 10^4^ 293FT-ACE2 or 2 × 10^4^ Huh-7 cells for 20 h. After incubation, the supernatant was removed and 1×Passive lysis buffer was added at 50 μL/well to lyse the cells. Relative light unit (RLU) of the cell lysate was measured to evaluate luciferase activity (LB960 Microplate Luminometer CentroXS3). Neutralization (%) = (Positive Control RLUs-Sample RLUs)/(Positive Control RLUs-Negative Control RLUs) × 100%. The nAb titer (50% inhibitory dilution, NAT_50_), which is defined as the serum dilution at which the RLUs were reduced by 50% when compared with the positive control wells, was calculated according to the Reed–Muench formula.

### 2.14. Enzyme-Linked Immune Absorbent Spot (ELISpot) Assay

RBD-Specific T-cell responses were quantified by using IFN-γ and IL4 ELISpot kit (Mabtech). Briefly, mouse splenocytes were isolated and cultured at a density of 2 × 10^5^ cells/100 μL per well into ELISpot plates. RBD peptide pools, with a concentration of 2 μg/mL in a volume of 100 μL per well, were added or not (for negative control wells). Assay plates were incubated overnight at 37 °C in a 5% CO_2_ incubator and developed using substrate until distinct spots emerged. The spots were counted in the enzyme-linked spot analyzer (ImmunoSpot^®^ S6, CTL). The number of IFN-γ and IL-4 secreting cells was obtained by subtracting the negative control number. Values below zero were presented as zero. The results were reported as spot-forming cells (SFCs) per million splenocytes. Data were analyzed by GraphPad Prism.

### 2.15. Opsonophagocytic Assay

The DMF-differentiated HL60 cells in Opsonization Buffer B (OBB) were suspended at 1 × 10^7^ cells/mL in Opsonization Buffer B (OBB). The OBB was added to a 96-well plate at 10 μL/well, and then the serum was serially diluted and added to a 96-well plate at 20μL/well. Then 20 μL of Pneumococcal Polysaccharide Powder Type 14 were added to incubate for 30 min at RT and room air on a mini-orbital shaker (600 rpm). After incubation, 50 μL HL60/complement mixture (4:1) was added and incubated on a mini-orbital shaker (600 rpm) for 45 min at 37 °C with 5% CO_2_. After the incubation, the 96-well plate was placed on ice for 20 min. 5 μL of reaction mixture was spotted from each well onto THYA plates, and the plates were tilted to shape the spots into a small strip of fluid (~2–3 cm long). The THYA plates were left at RT for 20 min to allow the excess fluid to seep into the agar. Following, 15 mL of overlay agar containing TTC was added to THYA plates. The plates were incubated at RT for 20 min. After the overlay had solidified, the THYA plates were incubated at 37 °C in 5% CO_2_ incubator for 16~18 h. After overnight incubation, the colonies (Colony Forming Unit, CFU) were counted using an automated counter and the data were analyzed. OPA titer was calculated according to the Reed–Muench formula and defined as the dilution of serum that kills 50% of the target bacteria.

### 2.16. Challenge Assay of Rhesus Macaques

Rhesus macaques (5–9 years old, n = 4/group) were immunized with three different doses of SCTV01A (1, 3, and 10 μg, respectively) formulated with 10 mg SCT-VA02B for three times via intramuscular injections on Day 0, Day 16, and Day 29. The control group was treated with an equal volume of SCT-VA02B alone. Serum samples were collected before vaccination and used in ELISA and viral neutralization assays. Six days after the third vaccination, 5.5 × 10^5^ PFU/mL of the SARS-COV-2 (GD108# strain) was inoculated to the monkeys via nasal drops (5.5 × 10^5^ PFU/monkey). Body weight and temperature were monitored and routinely recorded. Viral loads of throat swabs, nasal swabs, and anal swabs were detected at 3 days post infection (dpi) and 7 dpi by qRT-PCR assay. X-ray was conducted at 0, 3, 5, and 7 dpi. At 7 dpi, monkeys were dissected to detect the viral loads of the lung, trachea, lymph, and other tissues. Hematoxylin and eosin (H&E) staining was used to detect histopathological changes in lung tissues.

### 2.17. Micro-Neutralization Assay

Monkey serum samples collected from immunized animals were inactivated at 56 °C for 0.5 h and serially diluted two-fold with DMEM medium in 96-well plates for micro-neutralization assay [30]. The diluted serums were mixed with a virus (GD108#) suspension of 0.05 MOI in 96-well plates at a ratio of 1:1, followed by incubation at 37 °C and 5% CO_2_ for 1 h. Vero-E6 cells were then added to the serum–virus mixture, and the plates were incubated at 37 °C and 5% CO_2_ for 5 days. The cytopathic effect (CPE) of each well was counted under microscopes, and the neutralizing titer was calculated by the dilution number of 100% protective condition.

### 2.18. Sample Collection and Viral Nucleic Acid Extraction

Monkeys were anesthetized with ketamine before challenge or sampling. Nasal swabs, throat swabs, anal swabs, and blood were collected during anesthesia. The swab specimens were lysed with 800 µL Trizol, 400 µL of which was extracted and washed with 50 µL water to prepare the RNA template. RNA samples were stored at −80 °C for subsequent quantitative real-time PCR (qRT-PCR) analysis of viral load. When dissecting and sampling, about 50 mg of each lobe of lungs and other tissues were homogenized with 500 μL Trizol, 400 μL of which were extracted and washed with 50 μL of water to prepare the RNA template. RNA samples were stored at -80°C for subsequent qRT-PCR analysis of viral load.

### 2.19. qRT-PCR

qRT-PCR was used to determine the copies of viral genomic RNA (gRNA) in different tissues, throat swabs, anal swabs, and nasal swabs. The N specific primer and probe sequences are shown below, identical to sequences provided by WHO and China CDC. Forward primer: 5′-GGGGAACTTCTCCTGCTAGAAT-3′, Reverse primer: 5′-CAGACATTTTGCTCTCAAGCTG-3′, SARS-CoV-2 nucleocapsid (N) Probe: 5′-FAMTTGCTGCTGCTTGACAGATT-TAMRA-3′. The PCR reactions were performed in duplicate at 95 °C for 10 s; and subjected to 40 cycles of 95 °C for 5 s, 60 °C for 20 s. The virus loads were determined, according to the standard curve established by serial tenfold dilutions of SARS-CoV-2 RNA.

### 2.20. X-ray

At 0, 3, 5, and 7 dpi, the lungs of experimental animals were monitored by a mobile X-ray machine, focusing on early small patchy shadows and interstitial changes in the lungs, as well as multiple ground glass shadow, infiltration shadow, consolidation, and pleural effusion.

### 2.21. Hematoxylin and Eosin (H&E) Staining

The lung tissues were fixed with 4% paraformaldehyde and paraffin-embedded. The tissue sections (3 μm) were dewaxed, rehydrated, and regularly stained with H&E. Two pathologists were double-blinded to observe the histopathological changes and gross anatomy of the lung.

### 2.22. Safety Evaluation of SCTV01A in Nonhuman Primates

Rhesus monkeys were randomly assigned to 4 groups (n = 10/group, 5/gender/group) as follows: 2 groups of macaques were immunized by intramuscular injection with low (40 μg) or high (120 μg) doses, and another 2 groups of macaques were immunized with adjuvant and physiological saline (placebo). The animals were injected once a week for 4 doses, and convalescence for two weeks. Parameters evaluated in safety study included general clinical observations, body weight, food consumption, body temperature, electrocardiogram, blood pressure, blood oxygen saturation, ophthalmoscopic examinations, hematology, coagulation, clinical chemistry, urinalysis, lymphocyte subsets (CD3^+^, CD4^+^, CD8^+^, CD4^+^CD8^+^ and CD20^+^), cytokine (IFN-γ, TNF-α, IL-2, IL-4 and IL-6), C-reactive protein, serum complement (C3 and C4), the specific IgG antibody and nAb, organ weights, and macroscopic and microscopic examinations. The first 3 animals/gender/group in each group were euthanized on the third day after the last dose (Day 25), and the remaining animals were necropsied at the end of the 2-week recovery period (Day 36).

### 2.23. Statistical Analysis

Statistical analysis in this study was performed with GraphPad Prism (version 8.0.1, GraphPad Software, San Diego, CA, USA). Data concerning IgG antibody and pseudovirus (PsV) neutralization titers, which followed the skew normal distribution, were presented as geometric mean titer (GMT) ± standard deviation (SD) and were log transformed, resulting in a normal distribution of the data, and then analyzed by unpaired two-tailed Welch’s tests. Other data in this paper met the criteria of normal distribution, were presented as Mean ± SD, and were analyzed using unpaired, two-tailed Student’s *t* test. *p* < 0.05 was considered statistically significant.

## 3. Results

### 3.1. Design, Production and Characterization of SCTV01A

The SCTV01A polysaccharide–protein-conjugated nanoparticle vaccine was prepared as described previously [28] by chemically conjugating recombinant SARS-CoV-2 RBD-Fc and PPS14 (Figure 1a). Recombinant RBD-Fc protein, which linked SARS-CoV-2 virus spike receptor binding domain (RBD, 319–541) with a human IgG1 Fc, was expressed in Chinese hamster ovary (CHO) cells. The molecular size of recombinant glycosylated RBD-Fc protein was about 57 KDa confirmed by reduced gel electrophoresis with a band purity > 98.0% (Figure 1b). PPS14 of *S. pneumoniae* was produced and identified as described previously [28]. The chemical conjugation process of PPS14 and SARS-CoV-2 RBD-Fc protein was performed at an average ratio of 1:3 using the reductive amination method. The developed RBD-Fc/PPS14 conjugate (SCTV01A) had a protein/polysaccharide mass ratio of 0.7:1, as detected by ultraviolet (UV) and phenol-sulfuric acid (PHS) methods. Quantitative analysis of the SEC-MALS results showed that the purity of SCTV01A was >80% and the molecular weight was about 500 KDa (Figure 1c). The TEM representative data showed that SCTV01A particles were constituted of homogeneous sphere-shaped nanoparticles with diameters ranging from 80~120 nm (Figure 1d). Further biophysical characterizations revealed the diameter of SCTV01A to be 92 + 14 nm measured by Dynamic light scattering (DLS). The particle size of the two methods was basically the same (Figure 1e). SCTV01A binds to hACE2 at a similar affinity (*K*_D_ = 1.75 × 10^−9^ M) to that of the RBD-Fc protein (*K*_D_ = 1.74 × 10^−9^ M) (Figure 1f). The affinity results demonstrated that SCTV01A reserves specifically to hACE2 with high avidity and indicated that the key neutralizing RBD epitopes of SCTV01A were not disrupted by the PPS14 conjugation.

### 3.2. SCTV01A Augments RBD-Specific Antigenicity in Mice

Previous papers demonstrated that Fc fragment of human IgG in the RBD-based vaccine, RBD-Fc, can act as an important immunopotentiator to enhance the immunogenicity of RBD [31]. To evaluate whether the capsular polysaccharide conjugation could further augment RBD-Fc-specific antigenicity, we compared the immune responses induced by SCTV01A or unconjugated RBD-Fc antigen with the SCT-VA02B or Alum adjuvant in adult C57BL/6 mice. The mice were vaccinated intramuscularly (3 μg) three times at 14-day intervals, and the sera were collected at the time points as shown in Figure 2a. The geometric mean titer (GMT) of RBD-specific IgG in sera was detected with ELISA by coating RBD-his protein. The results showed that SCTV01A induced a higher RBD IgG titer than RBD-Fc, with 1.6–2.0 fold and 5.7–14.3 fold higher, respectively, when using SCT-VA02B or Alum as adjuvant at each time point (Figure 2b,c). To compare the nAb titers, the serum 50% neutralization titer (NAT_50_) against SARS-CoV-2 PsV infection was measured. Consistent with the results from RBD IgG titers, SCTV01A elicited a strong serum nAb response, with a GMT NAT_50_ 3.8–33.1 fold and 1.8–33.8 fold, respectively, higher than that induced by RBD-Fc elicited by the SCT-VA02B or Alum as adjuvant (Figure 2b,c). Similar results were obtained with a lower 1 μg dose vaccination (Appendix A). ELISpot results also showed that SCTV01A induced a similar frequency of splenic RBD-specific IFN-γ- and IL-4-SFCs as RBD-Fc (Figure 2d). Additionally, serum samples vaccinated with SCTV01A at 35 days after boosting vaccination were tested for NAT_50_ titer against the PsV of the Delta, BA.1, and BA.4/BA.5 variants. The results demonstrated that SCTV01A moderately decreased NAT_50_ titer against the Delta but greatly reduced NAT_50_ titers against the BA.1 and BA.4/BA.5 due to the Omicron containing a large number of mutations in RBD compared with the WT (Figure 2e). In addition, SCTV01A also induced high opsonophagocytic activity (OPA) against *S. pneumoniae* serotype 14 compared with that of the adjuvant control or RBD-Fc (Figure 2f). Collectively, these results indicate that the PPS14 conjugation not only could effectively enhance the capacity of the RBD-Fc vaccine to stimulate SARS-CoV-2 RBD-specific nAb but also showed potential benefit of prophylaxis against *S. pneumoniae* serotype 14-related pneumococcal disease.

### 3.3. SCTV01A Induced a Protective Immune Response in Nonhuman Primates

To evaluate the protective efficacy in nonhuman primates, rhesus macaques were immunized with three different doses of SCTV01A (1, 3, and 10 μg, respectively) three times via intramuscular injections on Day 0, Day 16, and Day 29. Six days after the third immunization, rhesus macaques were inoculated with 5.5 × 10^5^ PFU/mL SARS-CoV-2 (GD108# strain) by intranasal (Figure 3a).

RBD-specific IgG antibody reached a peak after the third immunization in a dose-dependent manner. The GMT in immunized serum of 1 µg, 3 µg, and 10 µg SCTV01A were 181019, 608874, and 861078 at 4 days after the third vaccination (Figure 3b left). Similarly, there was a dose-dependent increase in neutralizing activity as measured by PsV neutralization assay. The NAT_50_ titer in animals vaccinated with three doses SCTV01A were 414, 1111, and 1001 at 4 days after the third vaccination (Figure 3b middle). GMT of the 100% neutralizing titers against authentic virus were 215, 215, and 256 in animals vaccine with low, medium, and high dose of SCTV01A, respectively (Figure 3b right). Therefore, SCTV01A can stimulate potent RBD-specific IgG and neutralize titers in rhesus macaques.

After infection, the body temperature and weight fluctuations of the animals were within normal range (Appendix A). The levels of viral load at day 3 and day 7 post infection (dpi) were significantly lower in the three-dose group than in adjuvant group from both throat and nasal swab specimens. At 7 dpi, viral load of throat swab negatively correlated with the SCTV01A dose level (Figure 3c). With regard to anal sections, viral load of SCTV01A immunized animals was also significantly lower than that of the control animals (Figure 3c). At 7 dpi, all animals were euthanized and tissues were taken, and then viral load and histological changes in the lung tissues were determined. The sgRNA load in the lungs of rhesus macaques showed that only three lung lobes in each of the vaccine dose groups had detectable virus RNA, as compared with 14 respiratory tract tissues in adjuvant group (Figure 3d). Among lymph and other organs such as heart, liver, spleen, kidney, and muscle, none of the tissues had detectable RNA in vaccine animals as compared with 10 tissues of adjuvant immunized animals with viral load above 4.7 log10 copies/g (Appendix A). Pathological appearance of lung showed local bleeding spots and X-ray indicated increasing in lung texture and widening of bilateral hila in the adjuvant group, with no obvious bleeding and indicated X-ray features in the SCTV01A immunized lungs (Figure 4a and Appendix A). Pathological scores of lungs were graded for H&E staining according to grading criteria in Appendix A. The pathology in adjuvant lung showed medium or severe interstitial pneumonia with expansion parenchymal wall, hemorrhage, and immune cell infiltration in alveolar cavity (Figure 4b). Pathological scores of SCTV01A 1 µg, 3 µg, and 10 µg animals were all significantly lower than that of the adjuvant group, showing less hemorrhage and less inflammatory infiltration (Figure 4b,c). No evidence of immune-enhanced inflammatory disease was observed. Therefore, SCTV01A vaccination could effectively reduce lung inflammation after SARS-CoV-2 infection with neither antibody-dependent enhancement (ADE) nor vaccine-enhanced diseases (VED) phenomenon. These results indicated the efficient protective effect of SCTV01A in nonhuman primates.

### 3.4. SCTV01A Is Safe in Rhesus Macaques

We evaluated the safety of SCTV01A by testing the long-term toxicity in rhesus macaques. Rhesus monkeys were randomly assigned to four groups (n = 10/group, 5/gender/group) as follows: two groups of macaques were immunized by intramuscular injection with low (40 μg) or high (120 μg) dose, and another two groups were immunized with adjuvant or physiological saline (placebo). The animals were injected once a week for four doses, and convalescence for two weeks (Appendix A). The first three animals/gender/group in each group were euthanized on the third day after the last dose (Day 25), and the remaining animals were necropsied at the end of the 2-week recovery period (Day 36). No death or moribundity of dosing related was observed in animals of any groups throughout the study. No abnormalities in the gross anatomy of the euthanized animals in each dosed group on Day 25 and Day 36 were observed (data not shown). During the study, no vaccine related changes in body weight, body temperature, electrocardiogram, blood pressure, blood oxygen saturation, ophthalmoscopic examinations, hematology, clinical chemistry, urinalysis, lymphocyte subsets, and serum complement were observed in animals at all dose levels (Appendix A and data not shown). There are no differences between vaccine-inoculated animals and control animals in lymphocyte subgroup distribution (CD3^+^, CD4^+^, CD8^+^, CD4^+^CD8^+^, and CD20^+^, Figure 5a), cytokine profiles (IFN-γ, TNF-α, IL-2, IL-4, Figure 5b), and complements (C3, C4, data not shown). Compared to the placebo, statistically significant increases of IL-6 were noted in animals of SCTV01A low and high dose groups on Day 2. Increases of C Reaction Protein (CRP) were noted in adjuvant, low and high dose groups on Day 2 and Day 23 (data not shown). The changes above were considered to be related to the inflammatory responses at the injection site and/or immune response and could be fully recovered at the end of the 2-week recovery period.

In rhesus macaques, the titers of RBD-specific antibodies were markedly increased 7 days after the second immunization (Day 14) with low or high dose of SCTV01A compared with those in the control group and those in the vaccine group on Day 7. The RBD-specific antibody in SCTV01A vaccine group continued to arise at Day 21 and reached the peak at the end of the 2-week recovery period with the titer > 6 log (Appendix A). Similar to the SARS-CoV-2 RBD-specific IgG antibody response, the nAb levels were significantly increased at Day 14 and reached the peak at Day 36 with the titer > 1000 (Appendix A). As expected, in the control group the SARS-CoV-2 RBD-specific antibody and neutralizing activity were maintained at low levels throughout the study period.

## 4. Discussion

The COVID-19 pandemic caused by SARS-CoV-2 infection has led to a global health crisis. In order to prevent the transmission of SARS-CoV-2 around the world, vaccines with good efficacy and safety are urgently needed. SARS-CoV-2 infects human host cells through the binding of S proteins to cell surface-expressed ACE2 receptors [3,4]. To block virus infection, extracellular domain or the RBD of S protein is widely used as a candidate antigen for vaccine manufacturing [6,7,8]. Recombinant protein vaccines are of great importance for improved safety, high-yield production, and no need for low-temperature storage. Yet, the relatively low immunogenicity of protein vaccines indicated that it is necessary to reinforce the immune responses of protein vaccines through modification of antigens or with the use of a proper adjuvant.

The human IgG Fc fragment can serve as a vaccine adjuvant by promoting cellular and humoral immune responses. Various studies have demonstrated that Fc fragment of human IgG in the SARS-CoV-2 RBD-based vaccine, RBD-Fc, could enhance the immunogenicity of RBD [31,32,33]. Our study, for the first time, showed that SCTV01A, a candidate COVID-19 vaccine composed of PPS14-conjugated RBD-Fc fusion protein, could significantly reinforce the humoral immune response of RBD-Fc. In C57BL/6 mice, SCTV01A induced higher levels of RBD-specific antibodies and potent neutralizing activity against SARS-CoV-2, when compared with RBD-Fc whether formulated SCT-VA02B or Alum adjuvant (Figure 2b,c). The results indicated the importance of PPS14 conjugation to SCTV01A-induced immunity and were consistent with our previous conclusion in which conjugation of RBD proteins with PPS14 polysaccharides led to significantly enhanced immunogenicity [28]. Interestingly, vaccination with SCTV01A also induced high OPA in mice (Figure 2f) and was supposed to confer protection against *S. pneumoniae* infection [34,35]. Previous studies showed that the Fc fragment could facilitate the induction of a stronger T cell immune response in mice [36,37]. SCTV01A did not enhance T cell immune responses (Figure 2d), probably because the T cell immune response had reached saturation after fusion with Fc, as PPS14 enhanced RBD-specific T cell response in our previous mice immune study [28].

To determine whether the sera from immunized mice could neutralize the variants of concern (VOCs), we tested the neutralization activity of the sera against three pseudotyped SARS-CoV-2 variants: Delta, BA.1, and BA.4/BA.5. SCTV01A exhibited high nAb titers against the Delta variant, but reduced titers against Omicron variants which exhibited potent capabilities for immune evasion [38,39,40]. A significant decrease in neutralization titers against the Omicron variant was also reported for CoronaVac [41], NVX-CoV2373 [42], BNT162b2 [43], mRNA-1273 [44], and ChAdOx1 [45] vaccines which were designed based on the original strain. Thus, the second-generation vaccine with multicomponent or variant-matched vaccine may provide high protective efficacy against high-risk variants.

SCT-VA02B is a squalene-based oil-in-water emulsion similar to adjuvant systems 03 (AS03) and MF59^®^. Preclinical studies have proved that both adjuvant could enhance humoral and cellular immune responses and induce strong protection against SARS-CoV-2 challenge in RBD- or S-protein based vaccine [46,47]. SARS-CoV-2 recombinant protein vaccine with AS03 or MF59^®^ adjuvant also showed acceptable safety and reactogenicity, and robust immunogenicity in clinical trials [48,49,50]. The previously designed SCT-VA02B-adjuvanted polysaccharide–RBD-conjugated nanoparticle vaccine protected mice from SARS-CoV-2 challenge in murine models [28]. SCTV01A-formulated SCT-VA02B also demonstrated clear protective efficacy in a nonhuman primate challenge model. SCTV01A vaccination stimulated a high titer of RBD-specific IgG antibodies and neutralizing titer after three doses of vaccination in rhesus macaques (Figure 3b). SARS-CoV-2 challenge caused a broad infection of monkeys in different tissues including respiratory system lymph node, heart, spleen, kidney, and muscle whereas lung, trachea, and lymph node have the highest viral load. However, viral load in lung tissue of vaccinated animals was significantly reduced and that of trachea, lymph node, and other tissues are completely removed, suggesting that vaccination of SCTV01A prevents virus replication in the upper and lower respiratory tract and other tissues (Figure 3d and Appendix A). Importantly, we did not see any immune-enhanced diseases such as ADE and VED in vaccinated animals. These data suggest that SCTV01A elicited robust neutralizing activities and exhibited excellent protection efficacy against challenge by SARS-CoV-2.

Safety tests in animals are critical for vaccine development before it is applied to first-in-human clinical trials. The previously developed polysaccharide–RBD-conjugated nanoparticle vaccine has been proved to induce potent humoral and cellular immune responses in several animal models [28], but the safety and toxicity of this newly conjugated type of vaccine have not been verified. Nonhuman primates, such as macaques, are suitable for assessing vaccine immunogenicity and safety, as they mimic their human counterparts in multiple ways [51,52]. We next systematically evaluated the safety of SCTV01A formulated with SCT-VA02B in macaques by recording a number of clinical observations and biological indices. Following repeated intramuscular injection of SCTV01A to rhesus monkeys once a week for four doses, no systemic toxicity was observed in the animals at the dosage of 40 and 120 μg/animal during the period of administration and the end of 2-week recovery period. Vaccine-related changes were observed, such as an increase of IL-6 and CRP which were noted in the animals of adjuvant and vaccine groups at some time points. These transient responses that occurred may be related to inflammatory responses at the injection site and/or immune responses. In addition, the humoral immune response induced by the SCTV01A vaccine was evaluated. The serum anti-RBD antibody and PsV neutralization titers elicited by SCTV01A were significantly increased after two immunizations (Appendix A). Our results indicated that the SCTV01A vaccine was safe and well-tolerated in nonhuman primates; perhaps this conclusion could be extrapolated to the polysaccharide–RBD-conjugated nanoparticle vaccine [28]. The no observed adverse effect level (NOAEL) was 120 μg/animal in the repeated dose toxicity study on rhesus monkeys.

In conclusion, our data suggested that SCTV01A, by chemically conjugating recombinant SARS-CoV-2 RBD-Fc and PPS14, elicits the humoral immune response and production of potent nAbs in mice. SCTV01A provided protection against SARS-CoV-2 infection in a nonhuman primate challenge model. Furthermore, this vaccine is safe and well-tolerated, and it does not cause any adverse clinical symptoms or vaccine-related pathological changes in the rhesus macaques’ long-term toxicity study. Taken together, the existing immunological and toxicological evaluation results have demonstrated that SCTV01A is safe and effective, which will be of guiding significance in both the design of clinical trial protocol and future clinical application.

## Figures and Tables

**Figure 1 vaccines-11-00526-f001:**
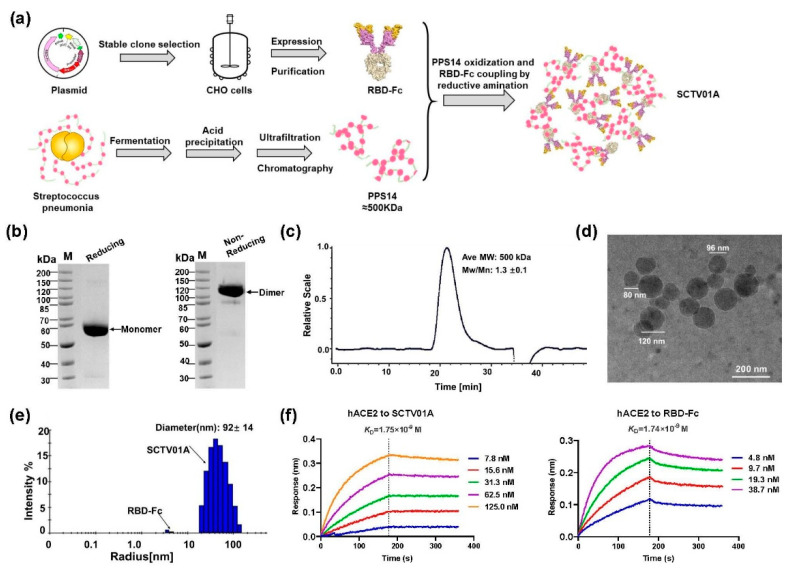
Design and characterization of SCTV01A. (**a**) Experimental design and workflow for the preparation of SCTV01A. The SARS-CoV-2 RBD-Fc was expressed in CHO cells. PPS14 was produced from *S. pneumoniae* by the acid precipitation method and conjugated with RBD-Fc utilizing the reductive amination method. (**b**) The molecular weight and purity of RBD-Fc were detected by NuPAGE. (**c**) The purity, molecular weight (MW), and polydispersity of SCTV01A were tested by SEC-MALS. (**d**) TEM of SCTV01A showed nanoparticles with heterogeneous diameters (scale bar = 200 nm). (**e**) The molecular diameter of SCTV01A and RBD-Fc was identified by DLS. (**f**) Binding affinity of SCTV01A and RBD-Fc to human ACE2 receptor using the BLI method.

**Figure 2 vaccines-11-00526-f002:**
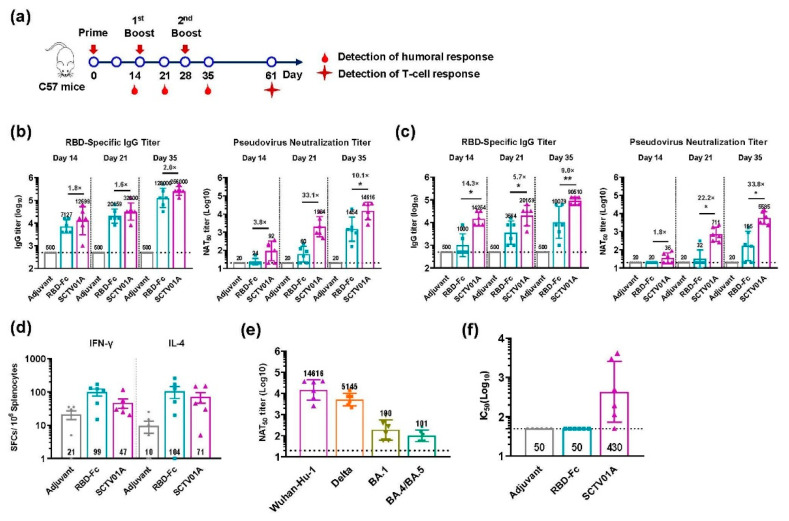
SCTV01A enhances RBD-Fc-specific immune responses in C57BL/6 mice. (**a**) Schematic diagram of immunization and sample collection. C57BL/6 mice (n = 6/group) were intramuscularly immunized with 3 μg of SCTV01A or RBD-Fc formulated with SCT-VA02B or Alum adjuvant three times with a 14-day interval. (**b**) RBD specific IgG titer and NAT_50_ titer formulated with SCT-VA02B. (**c**) RBD specific IgG titer and NAT_50_ titer formulated with Alum. (**d**) The frequencies of RBD-specific IFN-γ- and IL-4-SFCs in total splenocytes were determined by ELISpot formulated with SCT-VA02B. (**e**) NAT_50_ titer against the Delta, BA.1, and BA.4/BA.5 variants of SCTV01A formulated with SCT-VA02B at Day 35. (**f**) OPA of SCTV01A-immunized sera as determined by opsonophagocytic assay formulated with SCT-VA02B at Day 35. The GMT of RBD-specific antibody titer and NAT_50_ titer against the Wuhan-Hu-1 strain were determined by ELISA and SARS-CoV-2 PsV neutralization assay. The dashed black lines indicate the limit of detection (LOD). The LOD of the RBD-specific antibody ELISA is 500, and the LOD of the SARS-CoV-2 PsV neutralization assay is 20. For detection of the T-cell response, mice were euthanized at 61 days post first immunization, and their spleens were dissected for ELISpot. For (**b**,**c**), levels of RBD-specific IgG titers and NAT_50_ titers were compared across SCTV01A and RBD-Fc by unpaired two-tailed Welch’s tests. * *p* ≤ 0.05 and ** *p* ≤ 0.01 represent statistical significance. The unlabeled comparisons were not significant, with *p* ≥ 0.05.

**Figure 3 vaccines-11-00526-f003:**
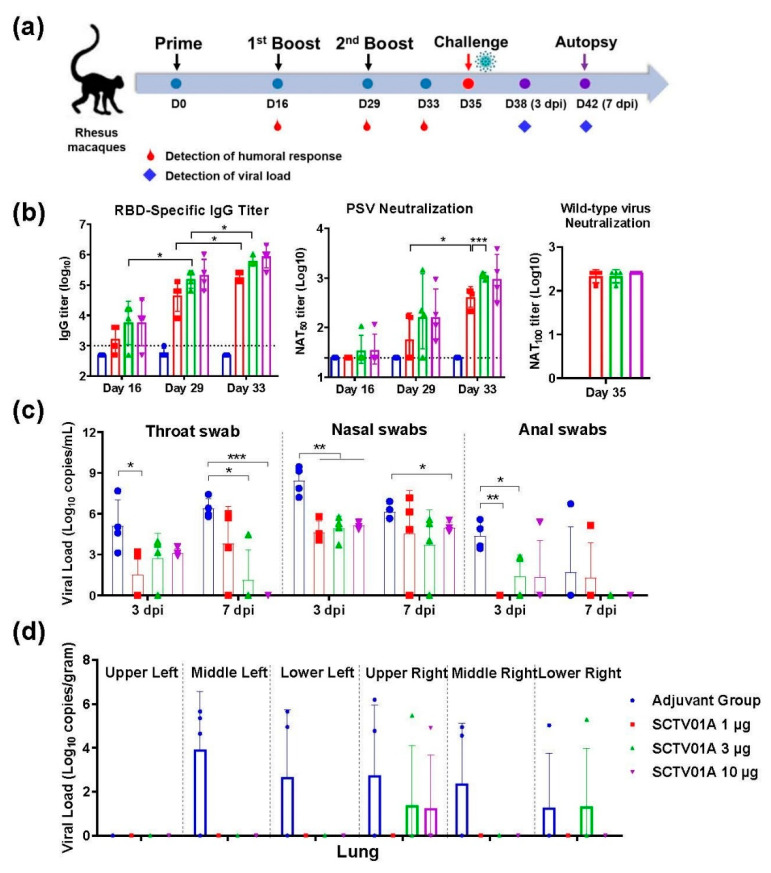
Immunogenicity and protective efficacy of SCTV01A in rhesus macaques. (**a**) Immunization and challenge procedures for rhesus macaques. Rhesus macaques (n = 4/group) were immunized with 1 μg, 3 μg, or 10 μg SCTV01A/SCT-VA02B on day 0, 16, and 29, respectively. The control group was treated with an equal volume of SCT-VA02B. The SARS-CoV-2 (Strain: GD108#) challenge was conducted on day 35. Swabs of throat, nasal, and anal were collected at 3 and 7 dpi. Macaques were sacrificed at 7 dpi. Tissues, such as lungs, heart, and liver, were collected. (**b**) RBD-specific IgG titer (left) and PsV NAT_50_ titer (middle) at the indicated time points post-immunization. SARS-CoV-2 authentic virus neutralizing titer NAT_100_ at Day35 (right). (**c**) Viral load of throat, nasal, and anal swabs at 3 dpi and 7 dpi. (**d**) SARS-CoV-2 viral load was determined using RT-qPCR at 7 dpi in lungs. Statistical analyses were performed using unpaired two-tailed Welch’s tests. * *p* < 0.05, ** *p* < 0.001, *** *p* < 0.0001.

**Figure 4 vaccines-11-00526-f004:**
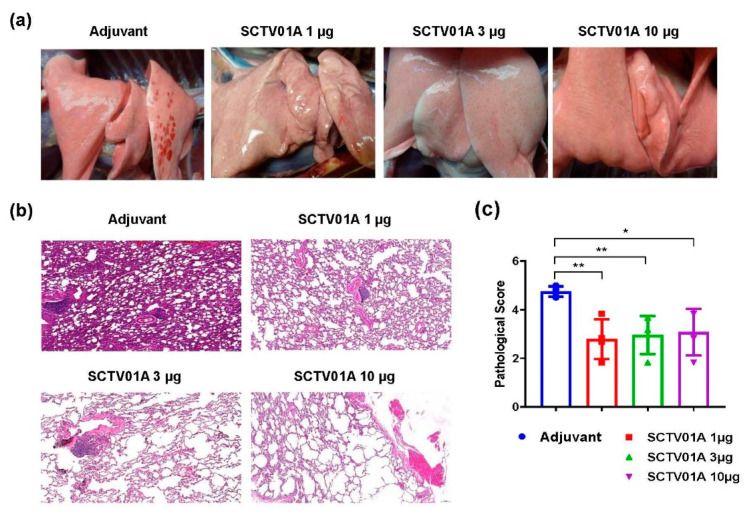
Lung pathology is reduced in SCTV01A-vaccinated rhesus macaques after SARS-CoV-2 challenge. (**a**) Representative gross anatomy of lung tissue of adjuvant, SCTV01A vaccines animals. (**b**) Representative histopathological staining of the lungs from macaques at 7 dpi. (**c**) Statistical pathological scores of lung in different vaccination groups. Statistical analyses were performed using unpaired two-tailed tests. * *p* < 0.05, ** *p* < 0.001.

**Figure 5 vaccines-11-00526-f005:**
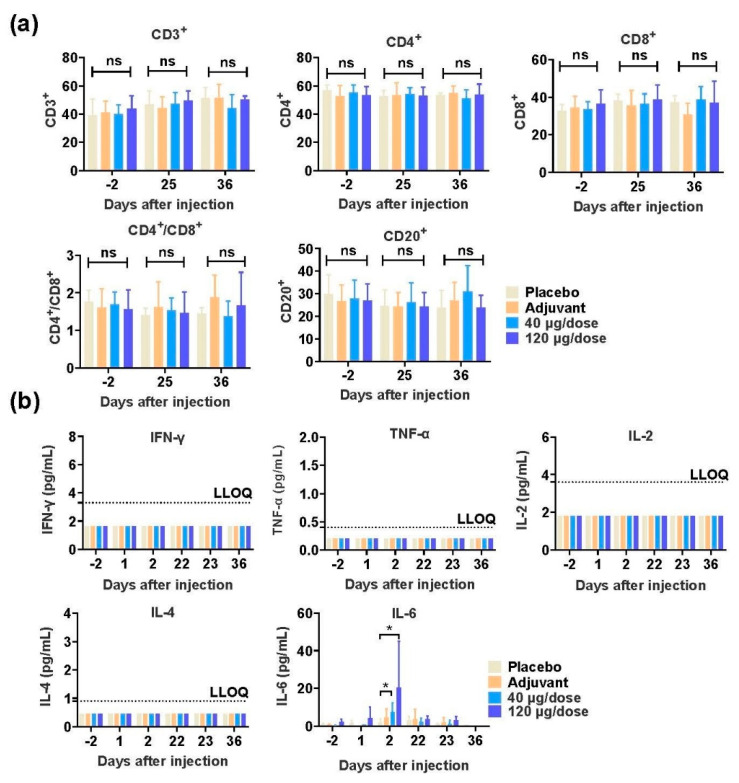
Safety evaluation of SCTV01A in nonhuman primates. Rhesus macaques (n = 10/group) were immunized four times at days 1, 8, 15, and 22 intramuscularly with low-dose (40 μg) or high-dose (120 μg) SCTV01A or adjuvant only or placebo. Hematological analysis in all four groups of macaques. (**a**) Percentage of lymphocytes, including CD3^+^, CD4^+^, CD8^+^, CD4^+^, CD8^+^, and CD20^+^, were monitored at days −2 (2 days before vaccination), 25 (3 days after the third vaccination), and 36 (14 days after the third vaccination). (**b**) Key cytokines containing IFN-γ, TNF-α, IL-2, IL-4, and IL-6 were examined at days −2, 25, and 36 after vaccination. Statistical analyses were performed using unpaired two-tailed tests. * *p* < 0.05.

## Data Availability

All data generated or analyzed during this study are included in this published article and its Appendix A.

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
