# Peer review of "A Polysaccharide-RBD-Fc-Conjugated COVID-19 Vaccine, SCTV01A, Showed High Immunogenicity and Low Toxicity in Animal Models"

_vaccines, 2023, doi:10.3390/vaccines11030526_

Round 1

Reviewer 1 Report

The manuscript entitled "A Polysaccharide-RBD-Fc-Conjugated COVID-19 Vaccine, SCTV01A, Showed High Immunogenicity and Low Toxicity in Animal Models" reports findings on the development of a COVID-19 vaccine composed of recombinant SARS-CoV-2 RBD-Fc conjugated to PPS14. The study was well-designed and presented well. However, there are minor issues to be addressed:

- In "2.4. Preparation of pneumococcal capsular polysaccharides", is there a strain or catalog number for the Streptococcus pneumoniae? The name of the bacterium should be written italic. Also, how was the purified PPS14 analyzed?

- It is better to give the method "2.14. Micro-neutralization assay" after immunization of monkeys.

- In "2.21. H&E staining", full name of H&E should be written.

- The results obtained in this study should be discussed with the ones obtained by the authors previously (Deng et al. Adv. Mater. 2022, 2200443).

- The text should be revised for the typos. The fonts should be uniform. The word "title" should be removed from the title. 

Author Response

- In "2.4. Preparation of pneumococcal capsular polysaccharides", is there a strain or catalog number for the Streptococcus pneumoniae? The name of the bacterium should be written italic. Also, how was the purified PPS14 analyzed?

Response: Thank you for your professional suggestion.

We added the isolate number for the Streptococcus pneumonia serotype 14 in the Materials and Methods section on page 3.

S. pneumoniae isolate CMCC 31614 (serotype 14) was obtained from the National Center for Medical Culture Collections (CMCC, China).”

The names of the bacteria have been changed to italic in our paper.

The purified PPS14 was analyzed by Nuclear magnetic resonance (NMR) as previously described (Deng et al. Adv. Mater. 2022, 2200443). Three lots of PPS14 showed identical signal peaks at the corresponding areas (Figure R1). Specific antibodies was also used to identify PPS14 (data not shown). We added a brief description of the identification in the paper as shown in below.

“PPS14 of S. pneumoniae was produced and identified as described previously [1].”

Figure R1. NMR spectrum of PPS14 saccharide residue components and structure analysis (3 lots): containing glucose and N-acetyl glucose groups with peaks at 4.88 ppm (parts per million) and galactose groups with peaks at 4.87 ppm. Peaks in the 4.59–3.50 ppm region were attributed to ring protons, and peaks in the 2.18 ppm region were attributed to N-acetyl groups.

- It is better to give the method "2.14. Micro-neutralization assay" after immunization of monkeys.

Response: We appreciate this suggestion. The “2.17. Micro-neutralization assay” instead of the “2.14. Micro-neutralization assay” has been adjusted after immunization of monkeys.

- In "2.21. H&E staining", full name of H&E should be written.

Response: Thanks for your careful review. The full name of H&E has been added. Please see page 7 “2.21. Hematoxylin and eosin (H&E) staining”.

- The results obtained in this study should be discussed with the ones obtained by the authors previously (Deng et al. Adv. Mater. 2022, 2200443).

Response: Thanks for your constructive suggestion. We have added a comparison with the polysaccharide–RBD-conjugated nanoparticle vaccine (Deng et al. Adv. Mater. 2022, 2200443) in the discussion section.

“The results indicated the importance of PPS14 conjugation to SCTV01A-induced immunity which is consistent with our previous conclusion that conjugation of RBD proteins with PPS14 polysaccharides led to significantly enhanced immunogenicity [1].”

“The previously designed SCT-VA02B-adjuvanted polysaccharide–RBD-conjugated nanoparticle vaccine protected mice from SARS-CoV-2 challenge in murine models [1].”

“Our results indicated that the SCTV01A vaccine was safe and well tolerated in nonhuman primates, perhaps this conclusion could be extrapolated to the polysaccharide–RBD-conjugated nanoparticle vaccine [1].”

- The text should be revised for the typos. The fonts should be uniform. The word "title" should be removed from the title.

Response: Thanks for the detailed suggestion. We changed some typos and unified the font size in our paper. The word “title” has been removed.

Reviewer 2 Report

The SARS-CoV-2 receptor-binding domain (RBD) is the protruding site of the Spike (S) protein that mediates viral-cell fusion during the initial infection event through binding of the human receptor angiotensin-converting enzyme 2 (hACE2). The RBD contains multiple neutralizing epitopes and several RBD-binding monoclonal antibodies (mAbs), exhibiting greater neutralizing potency against SARS-CoV-2 and SARS-CoV. Additionally, it has been established that RBD protein can induce both humoral immunity and T cell immune response in animals and exhibit protection against SARS-CoV-2 challenge.

Previous published research demonstrated that Fc fragment of human IgG in the RBD-based vaccine, RBD-Fc, can act as an important immunopotentiator to enhance the immunogenicity of RBD. This means that when antigens are linked to the Fc region of an immunoglobulin, the antigen processing and presentation can be improved through Fc-Fc.R-mediated antigen targeting to antigen presenting cells (APCs) as can also promote subsequent endocytosis.

Hitherto, it is well-known that S. pneumoniae pneumococcal polysaccharides (PPS) from S. pneumoniae serotype 14 (PPS14) showed a superior binding capacity to CLRs on DCs, and can be efficiently internalized by DCs. And, when a protein antigen is covalently conjugated to PS, the PS-mediated antigen crosslinking can induce stronger B-cell receptors (BCRs), cross-linking, antigen internalization, as well as presentation by APC.

In this manuscript, the authors have described an elegant synthesis of a novel nanoparticle vaccine, SCTV01A, prepared by chemically conjugating recombinant SARS-CoV-2 RBD-Fc  portion with the S. pneumoniae pneumococcal polysaccharides PPS14.

This novel nanoparticle vaccine SCTV01A was shown to  induce higher SARS-CoV-2 RBD-specific neutralizing antibodies (nAbs) than the RBD-Fc vaccine which protected nonhuman primates against SARS-CoV-2 infection. Likewise, SCTV01A showed well tolerance and did not cause any abnormal toxicity in nonhuman primates.

Therefore,SCTV01A may be a safe and effective vaccine to protect against SARS-CoV-2 and S. pneumoniae infection.

This manuscript is very well written and the immunological and toxicological evaluation results have demonstrated that SCTV01A is safe and effective and also indicate significant improvement in both vaccine design and future clinical applications against SARS-CoV-2 infection.

A minor correction is needed

1.The legend of Figure 2 is incomplete, add what (a), (b), (c), (d), (e) and (f) stand for!

The SARS-CoV-2 receptor-binding domain (RBD) is the protruding site of the Spike (S) protein that mediates viral-cell fusion during the initial infection event through binding of the human receptor angiotensin-converting enzyme 2 (hACE2). The RBD contains multiple neutralizing epitopes and several RBD-binding monoclonal antibodies (mAbs), exhibiting greater neutralizing potency against SARS-CoV-2 and SARS-CoV. Additionally, it has been established that RBD protein can induce both humoral immunity and T cell immune response in animals and exhibit protection against SARS-CoV-2 challenge.

Previous published research demonstrated that Fc fragment of human IgG in the RBD-based vaccine, RBD-Fc, can act as an important immunopotentiator to enhance the immunogenicity of RBD. This means that when antigens are linked to the Fc region of an immunoglobulin, the antigen processing and presentation can be improved through Fc-Fc.R-mediated antigen targeting to antigen presenting cells (APCs) as can also promote subsequent endocytosis.

Hitherto, it is well-known that S. pneumoniae pneumococcal polysaccharides (PPS) from S. pneumoniae serotype 14 (PPS14) showed a superior binding capacity to CLRs on DCs, and can be efficiently internalized by DCs. And, when a protein antigen is covalently conjugated to PS, the PS-mediated antigen crosslinking can induce stronger B-cell receptors (BCRs), cross-linking, antigen internalization, as well as presentation by APC.

In this manuscript, the authors have described an elegant synthesis of a novel nanoparticle vaccine, SCTV01A, prepared by chemically conjugating recombinant SARS-CoV-2 RBD-Fc  portion with the S. pneumoniae pneumococcal polysaccharides PPS14.

This novel nanoparticle vaccine SCTV01A was shown to  induce higher SARS-CoV-2 RBD-specific neutralizing antibodies (nAbs) than the RBD-Fc vaccine which protected nonhuman primates against SARS-CoV-2 infection. Likewise, SCTV01A showed well tolerance and did not cause any abnormal toxicity in nonhuman primates.

Therefore,SCTV01A may be a safe and effective vaccine to protect against SARS-CoV-2 and S. pneumoniae infection.

This manuscript is very well written and the immunological and toxicological evaluation results have demonstrated that SCTV01A is safe and effective and also indicate significant improvement in both vaccine design and future clinical applications against SARS-CoV-2 infection.

A minor correction is needed

1.The legend of Figure 2 is incomplete, add what (a), (b), (c), (d), (e) and (f) stand for!

The SARS-CoV-2 receptor-binding domain (RBD) is the protruding site of the Spike (S) protein that mediates viral-cell fusion during the initial infection event through binding of the human receptor angiotensin-converting enzyme 2 (hACE2). The RBD contains multiple neutralizing epitopes and several RBD-binding monoclonal antibodies (mAbs), exhibiting greater neutralizing potency against SARS-CoV-2 and SARS-CoV. Additionally, it has been established that RBD protein can induce both humoral immunity and T cell immune response in animals and exhibit protection against SARS-CoV-2 challenge.

Previous published research demonstrated that Fc fragment of human IgG in the RBD-based vaccine, RBD-Fc, can act as an important immunopotentiator to enhance the immunogenicity of RBD. This means that when antigens are linked to the Fc region of an immunoglobulin, the antigen processing and presentation can be improved through Fc-Fc.R-mediated antigen targeting to antigen presenting cells (APCs) as can also promote subsequent endocytosis.

Hitherto, it is well-known that S. pneumoniae pneumococcal polysaccharides (PPS) from S. pneumoniae serotype 14 (PPS14) showed a superior binding capacity to CLRs on DCs, and can be efficiently internalized by DCs. And, when a protein antigen is covalently conjugated to PS, the PS-mediated antigen crosslinking can induce stronger B-cell receptors (BCRs), cross-linking, antigen internalization, as well as presentation by APC.

In this manuscript, the authors have described an elegant synthesis of a novel nanoparticle vaccine, SCTV01A, prepared by chemically conjugating recombinant SARS-CoV-2 RBD-Fc  portion with the S. pneumoniae pneumococcal polysaccharides PPS14.

This novel nanoparticle vaccine SCTV01A was shown to  induce higher SARS-CoV-2 RBD-specific neutralizing antibodies (nAbs) than the RBD-Fc vaccine which protected nonhuman primates against SARS-CoV-2 infection. Likewise, SCTV01A showed well tolerance and did not cause any abnormal toxicity in nonhuman primates.

Therefore,SCTV01A may be a safe and effective vaccine to protect against SARS-CoV-2 and S. pneumoniae infection.

This manuscript is very well written and the immunological and toxicological evaluation results have demonstrated that SCTV01A is safe and effective and also indicate significant improvement in both vaccine design and future clinical applications against SARS-CoV-2 infection.

A minor correction is needed

1.The legend of Figure 2 is incomplete, add what (a), (b), (c), (d), (e) and (f) stand for!

The SARS-CoV-2 receptor-binding domain (RBD) is the protruding site of the Spike (S) protein that mediates viral-cell fusion during the initial infection event through binding of the human receptor angiotensin-converting enzyme 2 (hACE2). The RBD contains multiple neutralizing epitopes and several RBD-binding monoclonal antibodies (mAbs), exhibiting greater neutralizing potency against SARS-CoV-2 and SARS-CoV. Additionally, it has been established that RBD protein can induce both humoral immunity and T cell immune response in animals and exhibit protection against SARS-CoV-2 challenge.

Previous published research demonstrated that Fc fragment of human IgG in the RBD-based vaccine, RBD-Fc, can act as an important immunopotentiator to enhance the immunogenicity of RBD. This means that when antigens are linked to the Fc region of an immunoglobulin, the antigen processing and presentation can be improved through Fc-Fc.R-mediated antigen targeting to antigen presenting cells (APCs) as can also promote subsequent endocytosis.

Hitherto, it is well-known that S. pneumoniae pneumococcal polysaccharides (PPS) from S. pneumoniae serotype 14 (PPS14) showed a superior binding capacity to CLRs on DCs, and can be efficiently internalized by DCs. And, when a protein antigen is covalently conjugated to PS, the PS-mediated antigen crosslinking can induce stronger B-cell receptors (BCRs), cross-linking, antigen internalization, as well as presentation by APC.

In this manuscript, the authors have described an elegant synthesis of a novel nanoparticle vaccine, SCTV01A, prepared by chemically conjugating recombinant SARS-CoV-2 RBD-Fc  portion with the S. pneumoniae pneumococcal polysaccharides PPS14.

This novel nanoparticle vaccine SCTV01A was shown to  induce higher SARS-CoV-2 RBD-specific neutralizing antibodies (nAbs) than the RBD-Fc vaccine which protected nonhuman primates against SARS-CoV-2 infection. Likewise, SCTV01A showed well tolerance and did not cause any abnormal toxicity in nonhuman primates.

Therefore,SCTV01A may be a safe and effective vaccine to protect against SARS-CoV-2 and S. pneumoniae infection.

This manuscript is very well written and the immunological and toxicological evaluation results have demonstrated that SCTV01A is safe and effective and also indicate significant improvement in both vaccine design and future clinical applications against SARS-CoV-2 infection.

A minor correction is needed

1.The legend of Figure 2 is incomplete, add what (a), (b), (c), (d), (e) and (f) stand for!

The SARS-CoV-2 receptor-binding domain (RBD) is the protruding site of the Spike (S) protein that mediates viral-cell fusion during the initial infection event through binding of the human receptor angiotensin-converting enzyme 2 (hACE2). The RBD contains multiple neutralizing epitopes and several RBD-binding monoclonal antibodies (mAbs), exhibiting greater neutralizing potency against SARS-CoV-2 and SARS-CoV. Additionally, it has been established that RBD protein can induce both humoral immunity and T cell immune response in animals and exhibit protection against SARS-CoV-2 challenge.

Previous published research demonstrated that Fc fragment of human IgG in the RBD-based vaccine, RBD-Fc, can act as an important immunopotentiator to enhance the immunogenicity of RBD. This means that when antigens are linked to the Fc region of an immunoglobulin, the antigen processing and presentation can be improved through Fc-Fc.R-mediated antigen targeting to antigen presenting cells (APCs) as can also promote subsequent endocytosis.

Hitherto, it is well-known that S. pneumoniae pneumococcal polysaccharides (PPS) from S. pneumoniae serotype 14 (PPS14) showed a superior binding capacity to CLRs on DCs, and can be efficiently internalized by DCs. And, when a protein antigen is covalently conjugated to PS, the PS-mediated antigen crosslinking can induce stronger B-cell receptors (BCRs), cross-linking, antigen internalization, as well as presentation by APC.

In this manuscript, the authors have described an elegant synthesis of a novel nanoparticle vaccine, SCTV01A, prepared by chemically conjugating recombinant SARS-CoV-2 RBD-Fc  portion with the S. pneumoniae pneumococcal polysaccharides PPS14.

This novel nanoparticle vaccine SCTV01A was shown to  induce higher SARS-CoV-2 RBD-specific neutralizing antibodies (nAbs) than the RBD-Fc vaccine which protected nonhuman primates against SARS-CoV-2 infection. Likewise, SCTV01A showed well tolerance and did not cause any abnormal toxicity in nonhuman primates.

Therefore,SCTV01A may be a safe and effective vaccine to protect against SARS-CoV-2 and S. pneumoniae infection.

This manuscript is very well written and the immunological and toxicological evaluation results have demonstrated that SCTV01A is safe and effective and also indicate significant improvement in both vaccine design and future clinical applications against SARS-CoV-2 infection.

A minor correction is needed

1.The legend of Figure 2 is incomplete, add what (a), (b), (c), (d), (e) and (f) stand for!

The SARS-CoV-2 receptor-binding domain (RBD) is the protruding site of the Spike (S) protein that mediates viral-cell fusion during the initial infection event through binding of the human receptor angiotensin-converting enzyme 2 (hACE2). The RBD contains multiple neutralizing epitopes and several RBD-binding monoclonal antibodies (mAbs), exhibiting greater neutralizing potency against SARS-CoV-2 and SARS-CoV. Additionally, it has been established that RBD protein can induce both humoral immunity and T cell immune response in animals and exhibit protection against SARS-CoV-2 challenge.

Previous published research demonstrated that Fc fragment of human IgG in the RBD-based vaccine, RBD-Fc, can act as an important immunopotentiator to enhance the immunogenicity of RBD. This means that when antigens are linked to the Fc region of an immunoglobulin, the antigen processing and presentation can be improved through Fc-Fc.R-mediated antigen targeting to antigen presenting cells (APCs) as can also promote subsequent endocytosis.

Hitherto, it is well-known that S. pneumoniae pneumococcal polysaccharides (PPS) from S. pneumoniae serotype 14 (PPS14) showed a superior binding capacity to CLRs on DCs, and can be efficiently internalized by DCs. And, when a protein antigen is covalently conjugated to PS, the PS-mediated antigen crosslinking can induce stronger B-cell receptors (BCRs), cross-linking, antigen internalization, as well as presentation by APC.

In this manuscript, the authors have described an elegant synthesis of a novel nanoparticle vaccine, SCTV01A, prepared by chemically conjugating recombinant SARS-CoV-2 RBD-Fc  portion with the S. pneumoniae pneumococcal polysaccharides PPS14.

This novel nanoparticle vaccine SCTV01A was shown to  induce higher SARS-CoV-2 RBD-specific neutralizing antibodies (nAbs) than the RBD-Fc vaccine which protected nonhuman primates against SARS-CoV-2 infection. Likewise, SCTV01A showed well tolerance and did not cause any abnormal toxicity in nonhuman primates.

Therefore,SCTV01A may be a safe and effective vaccine to protect against SARS-CoV-2 and S. pneumoniae infection.

This manuscript is very well written and the immunological and toxicological evaluation results have demonstrated that SCTV01A is safe and effective and also indicate significant improvement in both vaccine design and future clinical applications against SARS-CoV-2 infection.

A minor correction is needed

1.The legend of Figure 2 is incomplete, add what (a), (b), (c), (d), (e) and (f) stand for!

The SARS-CoV-2 receptor-binding domain (RBD) is the protruding site of the Spike (S) protein that mediates viral-cell fusion during the initial infection event through binding of the human receptor angiotensin-converting enzyme 2 (hACE2). The RBD contains multiple neutralizing epitopes and several RBD-binding monoclonal antibodies (mAbs), exhibiting greater neutralizing potency against SARS-CoV-2 and SARS-CoV. Additionally, it has been established that RBD protein can induce both humoral immunity and T cell immune response in animals and exhibit protection against SARS-CoV-2 challenge.

Previous published research demonstrated that Fc fragment of human IgG in the RBD-based vaccine, RBD-Fc, can act as an important immunopotentiator to enhance the immunogenicity of RBD. This means that when antigens are linked to the Fc region of an immunoglobulin, the antigen processing and presentation can be improved through Fc-Fc.R-mediated antigen targeting to antigen presenting cells (APCs) as can also promote subsequent endocytosis.

Hitherto, it is well-known that S. pneumoniae pneumococcal polysaccharides (PPS) from S. pneumoniae serotype 14 (PPS14) showed a superior binding capacity to CLRs on DCs, and can be efficiently internalized by DCs. And, when a protein antigen is covalently conjugated to PS, the PS-mediated antigen crosslinking can induce stronger B-cell receptors (BCRs), cross-linking, antigen internalization, as well as presentation by APC.

In this manuscript, the authors have described an elegant synthesis of a novel nanoparticle vaccine, SCTV01A, prepared by chemically conjugating recombinant SARS-CoV-2 RBD-Fc  portion with the S. pneumoniae pneumococcal polysaccharides PPS14.

This novel nanoparticle vaccine SCTV01A was shown to  induce higher SARS-CoV-2 RBD-specific neutralizing antibodies (nAbs) than the RBD-Fc vaccine which protected nonhuman primates against SARS-CoV-2 infection. Likewise, SCTV01A showed well tolerance and did not cause any abnormal toxicity in nonhuman primates.

Therefore,SCTV01A may be a safe and effective vaccine to protect against SARS-CoV-2 and S. pneumoniae infection.

This manuscript is very well written and the immunological and toxicological evaluation results have demonstrated that SCTV01A is safe and effective and also indicate significant improvement in both vaccine design and future clinical applications against SARS-CoV-2 infection.

A minor correction is needed

1.The legend of Figure 2 is incomplete, add what (a), (b), (c), (d), (e) and (f) stand for!

Author Response

-The legend of Figure 2 is incomplete, add what (a), (b), (c), (d), (e) and (f) stand for!

Response: Thanks for your careful review. Each subfigure has been added an explanation in figure 2. Please let us know if you have other suggestions.

Figure 2. SCTV01A enhances RBD-Fc-specific immune responses in C57BL/6 mice. (a) Schematic diagram of immunization and sample collection. C57BL/6 mice (n=6/group) were intramuscularly immunized with 3 μg of SCTV01A or RBD-Fc formulated with SCT-VA02B or Alum adjuvant three times with a 14 days interval. (b) RBD specific IgG titer and NAT50 titer formulated with SCT-VA02B. (c) RBD specific IgG titer and NAT50 titer formulated with Alum. (d) The frequencies of RBD-specific IFN-γ- and IL-4-SFCs in total splenocytes were determined by ELISpot formulated with SCT-VA02B. (e) NAT50 titer against the Delta, BA.1 and BA.4/BA.5 variants of SCTV01A formulated with SCT-VA02B at Day 35. (f) OPA of SCTV01A-immunized sera as determined by opsonophagocytic assay formulated with SCT-VA02B at Day 35. The GMT of RBD-specific antibody titer and NAT50 titer against the Wuhan-Hu-1 strain were determined by ELISA and SARS-CoV-2 PsV neutralization assay. The dashed black lines indicate the limit of detection (LOD). The LOD of the RBD-specific antibody ELISA is 500, and the LOD of the SARS-CoV-2 PsV neutralization assay is 20. For detection of the T-cell response, mice were euthanized at 61 days post first immunization, and their spleens were dissected for ELISpot. For (b) and (c), levels of RBD-specific IgG titers and NAT50 titers were compared across SCTV01A and RBD-Fc by unpaired two-tailed Welch’s tests. *p ≤ 0.05 and **p ≤ 0.01 represent statistical significance. The unlabeled comparisons were not significant, with p ≥ 0.05.

Reviewer 3 Report

Title A Polysaccharide-RBD-Fc-Conjugated COVID-19 Vaccine, SCTV01A, Showed High Immunogenicity and Low Toxicity in Animal Models

Manuscript ID: vaccines-2202205

The authors have describe the development of a COVID-19 vaccine SCTV01A, by chemically conjugating recombinant SARS-CoV-2 RBD-Fc and PPS14  (Streptococcus pneumoniae serotype type 14 capsular polysaccharide). The PPS14 conjugation enhanced the immunogenicity of RBD-Fc in C57BL/6 mice whether formulated with SCT-VA02B or Alum adjuvant. SCTV01A also induced high opsonophagocytic activity (OPA) against S. pneumoniae serotype 14. In addition, SCTV01A stimulated potent neutralizing titers in rhesus macaques and effectively reduced lung inflammation after SARS-CoV-2 infection with neither antibody-dependent enhancement (ADE) nor vaccine-enhanced diseases (VED) phenomenon. Importantly, the long-term toxicity study of SCTV01A in rhesus macaques did not cause any abnormal toxicity and was tolerated at the highest tested dose (120 μg). The existing immunogenicity and toxicological evaluation results have demonstrated the safety and efficacy of SCTV01A, which will be a promising and feasible vaccine to protect against SARS-CoV-2 infection.

This vaccine candidate is highly innovative and the article describes its development and validation perfectly. The study design and its presentation are neat and good.  The results are also presented and explained in a perfectly understandable manner. 

The article can be accepted in its present form. THis vaccine candidate can be highly promising, as it can be a vaccine against pneumonia along with COVID-19.

Please correct the following 

Minor changes:

1. Section 2.16. Opsonophagocytic assay, correct the font

Author Response

-Section 2.16. Opsonophagocytic assay, correct the font.

Response: Thanks for the detailed suggestion. The font size has been corrected.
